# Associations between women's empowerment and children's health status in Ethiopia

**Solomon Kibret Abreha**[1]*, **Solomon Zena Walelign**[2,3], **Yacob Abrehe Zereyesus**[4]

**1** Department of Economics, University of Insubria, Varese, Italy, **2** Department of Geography, Norwegian University of Science and Technology, Trondheim, Norway, **3** School of Economics, University of Gondar, Gondar, Ethiopia, **4** Department of Agricultural Economics, Kansas State University, Manhattan, NY, United States of America

* abegenik.09@gmail.com

## Abstract

Although women's empowerment has gained attention over the last two decades, our understanding of the associations between different dimensions of women's empowerment and different children's health outcomes is limited. This study aims to measure the extent of women's empowerment and to examine its associations with the children's health status in Ethiopia. Data were obtained from the 2016 Ethiopian Demographic and Health Survey (EDHS). The sample is restricted to a sub-sample of 10,641 women from 15 to 49 years old and their children under the age of five years. We used children's height-for-age and weight-for-height Z-scores and pneumonia and anemia experience as indicators of children's health outcome. Women's empowerment is measured by five indices reflecting their participation in decision-making, attitudes towards wife-beating by husband, barriers to health care access, asset ownership, and socio-economic variables. These indicators of empowerment were constructed using exploratory and confirmatory factor analysis. A Multiple Indicators Multiple Causes (MIMIC) model was employed to examine the relationship between women's empowerment and latent child health outcomes, after controlling for relevant covariates. Results suggests that enhancing women's empowerment in the household in terms of their socio-economic status (i.e., increasing women's access to education, information, media, and promoting saving) was associated with less likelihood of the children's being stunted or wasted (p<0.05). Higher women's empowerment in terms of household decision-making power were also associated with better children's health status measured by the children's experience of pneumonia and anemia (p<0.05). All aspects of women's empowerment are not related with children's health indicators. Women's empowerment dimensions related with child health have a varying degree of association with the different children's health indicators. Gender-specific policies focusing on increasing women's access to education, media, information, and promoting saving and their participation in the household decision making are some of the strategies for improving their children's health and wellbeing.

**Data Availability Statement:** All DHS data are available from the DHS database: https://dhsprogram.com/what-we-do/survey/survey-display-478.cfm.

**Funding:** The author(s) received no specific funding for this work.

**Competing interests:** The authors have declared that no competing interests exist.

## 1. Introduction

Ensuring better child health outcomes remains a global public health challenge due to the prevalence of children's vulnerability to malnutrition and exposure to infectious diseases, particularly in developing countries [1,2]. Health outcomes during childhood, which is an essential period in human development, affects child's lifelong health and well-being [2]. Hence, child health has featured in the international development agenda of the last 20 years. Best examples for this are the adoption of the Millennium Development Goals (MDGs) and Sustainable Development Goals (SDGs) by the United Nations in 2000 and 2015, respectively. Both MDGs and SDGs have been dedicated to promoting healthy lives and well-being for all children, particularly MDG-4 which aims to reduce child mortality and SDG-3 which aims to end preventable deaths of newborns and under-5 children by 2030 [3,4].

Although progress has been made in reducing child mortality rate over the past two decades worldwide due to interventions aiming to meet the targets set by the MDGs and SDGs, an estimated 6.3 million children under the age of 15 years died in 2017 [5]. About 5.4 million of them were under the age of 5 and 2.5 million of those children died within the first month of life, equivalent to 15,000 under-five deaths per day [5]. Major disparity exists in under-5 child mortality across regions and countries: it varies from 316 deaths per 1,000 live births in Sierra Leone to 3–5 deaths per 1,000 live births in Iceland, Finland, and Japan [6]. Sub-Saharan Africa (SSA) remains the region with the highest under-5 mortality rate in the world in 2017 accounting for half of all deaths under 5 years. In SSA, 1 out of 13 dies before the child's fifth birthday, 14 times higher than in high-income countries, where in high-income countries, that number was 1 in 185 [5].

Worldwide in 2016, three quarters of children and young adolescents aged between 0 and 14 died due to complications from communicable, prenatal and nutrition specific conditions [5]. Childhood stunting and wasting remain among the most serious health problems in developing countries. Nearly half of all deaths in children under 5 are attributed to undernutrition [5]. Although the prevalence of stunting among children under 5 declined from 32.5% (198.2 million) in 2000 to 21.9% (149 million) in 2018 globally, only minor improvements have been achieved in some of the poorest regions of the world, especially South Asia and sub-Saharan Africa [7,8]. Stunting affected nearly two out of five children in South Asia while another two out of five children in sub-Saharan Africa in 2018 [8]. Stunting affects one-third of children under age 5 years in developing countries contributing to 14% of childhood deaths in these countries [8]. In 2018, three regions (South Asia, Eastern and Southern Africa and West and Central Africa) experienced extremely high rates of stunting with one third of children affected [8]. Wasting also affects the lives of children around the world. In 2018, a joint child malnutrition estimation by the UNICEF/WHO/World Bank revealed that, globally, over 49 million children under five years were wasted of which nearly 17 million were severely wasted [9]. Half of these wasted children lived in South Asia and 25% lived in sub-Saharan Africa. Moreover, pneumonia accounts for about 16 percent of the leading causes of death among children under age five [5].

Studies indicated that children's health is determined by many factors including family income, family employment and educational status, parental health condition and marital status, number of children in the family, gender and race of the child, whether the child has a health insurance and personal health care professional, environmental factors (e.g., traffic load at the place of residence, perceived environmental quality) and, community-level factors (e.g., availability of services in the community) (see e.g., [10–14]. Women, as the main caregivers of children in rural communities in developing countries, are particularly responsible for the nutrition and health-related decisions that impact the children in their care [15,16]. Given

such roles by women as the primary caregivers of children, it is reasonable to assert that their empowerment would impact the health status of their children. Scores of studies have in general documented that women's empowerment is critical to achieving development objectives and contributes to better family health outcomes including children's health [17,18]. Recent studies in Ethiopia have also acknowledged the significance of women's empowerment as one of the determinants of child health [19–23]. These studies provide evidence that women's empowerment (measured in terms of decision-making power, control over resources, autonomy) improved maternal and children's health.

However, two major limitations underpin the previous studies. First, most of the studies emphasize fewer dimensions of women empowerment (mainly decision-making power [24–29]), even though gender empowerment is multidimensional [20,30,31]. Second, studies on the assessment of women's empowerment and children's health focused on children's anthropometric indicators (height-for-age, weight-for-height, and weight-for-age) (see e.g. [16,21–23,32,33]), amid ample evidence that non-anthropometric related diseases are the major health problems and causes of deaths in developing countries [5,10,14,22,34,35]. By taking advantage of the 2016 Ethiopian Demographic and Health Survey (EDHS) dataset, the current study contributes to the child health and gender empowerment literature by assessing the association of multiple dimensions of women's empowerment with both anthropometric and non-anthropometric (e.g. anemia and pneumonia) child health measures.

This study, therefore, aims to investigate the association between multiple dimensions of women's empowerment (women's participation in household decision making, attitudes towards wife-beating, access to health care, education, information, and asset ownership) and multiple child health outcomes (exposure to anemia, pneumonia, wasting and stunting). Using data from the 2016 EDHS, we first constructed the five dimensions of women's empowerment employing exploratory and confirmatory factor analyses. We then applied Multiple Indicators Multiple Causes (MIMIC) to model the association between women's empowerment dimensions and children's health outcome, controlling for other covariates.

The remainder of the paper is organized as follows. Section 2 presents an overview of women's empowerment and children's health in Ethiopia. Section 3 presents the conceptual framework regarding women's empowerment and children's health. Section 4 provides description of the data and methods. Section 5 presents the results of the study. Section 6 discusses and section 7 concludes.

## 2. Women's empowerment and child health in Ethiopia

Ethiopia is the second-most populous landlocked country located in the Northeastern part of Africa with an estimated total population of about 105 Million in 2017 [36]. Ethiopia has one of the world's highest rates of child mortality in the world. Every year, more than 257,000 children under the age of five die and 120,000 of them die during neonatal period [37]. Although progress has been made over the last decade, child death is prevalent, the most common one being new-born death within the first 28 days of life (neonatal age) [38]. Based on [5] estimate the under-five mortality rate in Ethiopia in 1990 was 202 deaths per 1000 live births. In 2017 this figure is reduced to 59 deaths per 1000 live births. Pneumonia, diarrhea, and malaria, taken together with newborn deaths, account for most deaths among children under 5 years of age [39]. These deaths tend to occur with the poorest and most disadvantaged populations. Despite improvements in the reduction of the number of stunted and wasted children over the last 15 years in Ethiopia, under-nutrition is still a major risk factor contributing to the mortality and disease burden among children aged under 5 years. Undernutrition accounts for about 25% of child deaths through complications related from stunting and wasting [38]. Studies

indicate that parent's socio-economic status, mother's education level, geographical location and access to health care are among the major social determinants of under-five children's health in Ethiopia [39]. Gizaw M. [39] demonstrated that under-five mortality is prevalent among males, rural residents, and poor families. This study showed children born from uneducated mothers and children living in poor and marginalized regions were subjected to the highest risk of under-five deaths.

Women represent the majority of the population in Ethiopia (about 50.2%) [36]. However, Ethiopia has one of the lowest gender equality performance indicators: The 2010 Global Gender Gap report ranks Ethiopia at 121 out of 134 countries in terms of the magnitude and scope of gender disparities. Contrary to Ethiopia's progress in achieving many of the Millennium Development Goals (MDGs), improvement in women's empowerment (MDG-3) targets is minimal [40]. Women are strongly disadvantaged compared to men in several areas. They face restrictions in making decisions on matters of their interest and are disproportionately excluded from social, economic and political spheres [40]. The 2016 EDHS report revealed that only 48% of currently married women age 15–49 were employed in the 12 months before the survey, compared with 99% of currently married men in the same age group. In terms of earning, most women earn less than their husbands (about 58%) while a few women (about 16%) earn more than their husbands [41]. Women also experience a high rate of drop out from schooling after marriage– 25% of women were attending school at the time their first marriage, and the majority (75%) of these women stopped going to school after they married [41]. This affects women's educational attainment compared to men.

Previous studies on women's empowerment and children's health in Ethiopia have primarily focused on children's anthropometric indicators (height-for-age, weight-for-height, and weight-for-age) to represent child health outcomes (See e.g. [21–23]. Using Demographic and Health Surveys (2011–2016) data in five African countries (Ethiopia, Kenya, Rwanda, Tanzania, and Uganda), a recent study by Jones et al. [22], examined the pathways by which women's empowerment influences child nutritional status. The authors constructed three dimensions of women's empowerment namely "social/human assets, "intrinsic agency" (attitudes about intimate partner violence), and "instrumental agency" (influence in household decision making). They applied structural equation models to estimate the direct and indirect associations of women's empowerment dimensions with the three child nutritional status indicators represented by anaemia, height-for-age z-score (HAZ), and weight-for-age z-score (WHZ), while using women's BMI as a mediating factor. Kuche et al. [21] also assessed whether sociodemographic, agricultural diversity, and women's empowerment factors were associated with child dietary diversity and length-for-age z-score (LAZ) in children 6–23 months using data from Ethiopia. Using primary data from a baseline survey conducted for the evaluation of the Feed the Future (FtF) program in Ethiopia in 2013, Yimer [23] examined the impact of women's empowerment in agriculture on the nutrition outcomes of children and women. By applying a multivariate regression methods and instrumental variable techniques, they found all women's empowerment indicators were positively related to better dietary diversity for both children and women. Other previous studies in Ethiopia focused on the association of women's empowerment with infant mortality [28,42], children's immunization status [43], and maternal healthcare services [44,45].

Most of the aforementioned studies find that women's decision-making power in the household is positively associated with better nutritional status and mortality [21,22]. For instance, Kuche et al. [21] found positive associations between children dietary diversity and women's empowerment measured as women's household decision making power. Jone et al. [22] also found that women's empowerment dimensions (assets, intrinsic agency and instrumental agency) were indirectly and positively associated with child Height for Age Z-score

(HAZ) and child Weight for Age Z-score (WAZ) mediated through maternal BMI. Form the non-anthropometric child health indicator, Ebot et al, [43] found that in Ethiopia women participated in more household decisions are more likely to have fully vaccinated children. Fantahun et al. [42] and Alemayehu et al. [28] found that in Ethiopia women's empowerment is inversely associated with infant mortality.

## 3. Conceptual framework

Previous studies documented that women's empowerment is critical to achieving development objectives and contributes to better family health outcomes including child health [17,18]. Women's better access to health, education, employment, increased political participation, and control of resources (including house and land) are crucial for promoting sustainable development and improving health outcomes. Greater women's empowerment improves health and quality of life for women and their family members in two ways [19–21,46–48]. First, women who have higher decision making power are more likely to access health services and have control over health resources which improves child health outcomes [20]. Second, more empowered women are more likely to have fewer children and the children receive better childcare at home which in turn helps to improve child health outcomes [49].

The relationship between women's empowerment and children's health is described with Multiple Indicators Multiple Causes (MIMIC) framework a shown in Fig 1. Child health is conceptualized as an outcome variable that depends on the mother's empowerment and other socioeconomic variables. The MIMIC model is a type of Structural Equation Modelling (SEM) approach, which is useful when multiple dependent variables need to be tied together with multiple independent variables through an underlying latent variable [16]. The equations in the MIMIC model contain both observed and unobserved or latent random variables [50]. An attractive advantage of the MIMIC model is that it controls for error in measurement [16,50]. This approach has been applied in recent studies on women's empowerment and health outcomes in different contexts [16,19]. Therefore, we employ a MIMIC model to link observed anthropometric (i.e. child's wasting and stunting status) and non-anthropometric indicators (i.e. child's anemic status and exposure to pneumonia) and women's empowerment dimensions through latent children's health outcomes.

The structure of the MIMIC model can be represented by the path diagram as shown in Fig 1 [51]. Path diagram is a graphical representation of the SEM that shows a causal relationship between a latent variable and a set of exogenous factors. Oval shapes denote latent variables to be generated by the model whereas rectangles denote exogenous observed variables. The circles represent error terms. The MIMIC model consists of two parts: the measurement model which defines the relations between the two latent variables (i.e., anthropometric and non-anthropometric) and its observed variables (indicators) and the structural model which displays the causal link among the latent variables and the observed causes.

As shown in the measurement part of the MIMIC model of Fig 1, the observed child health indicators of stunting (child's height to age z-scores) and wasting (child's weight to height z-scores) reflecting the underlying latent child health status (hereafter, we named "anthropometric"). Similarly, anemia and pneumonia are the indicators for second latent child health outcome (hereafter, "non-anthropometric"). We separated the two latent health outcomes for two reasons. First, to show the outcomes comes from different category of health indicators i.e. one is from the children's anthropometric indicators (height-for-age z-score and weight-for-height z-score) and the other is from other types of child health outcomes (Pneumonia and Anemia). The second reason is, we need to compare the association of women's empowerment on these separate latent health outcomes and to see their differences. Hence, we aim to compare the

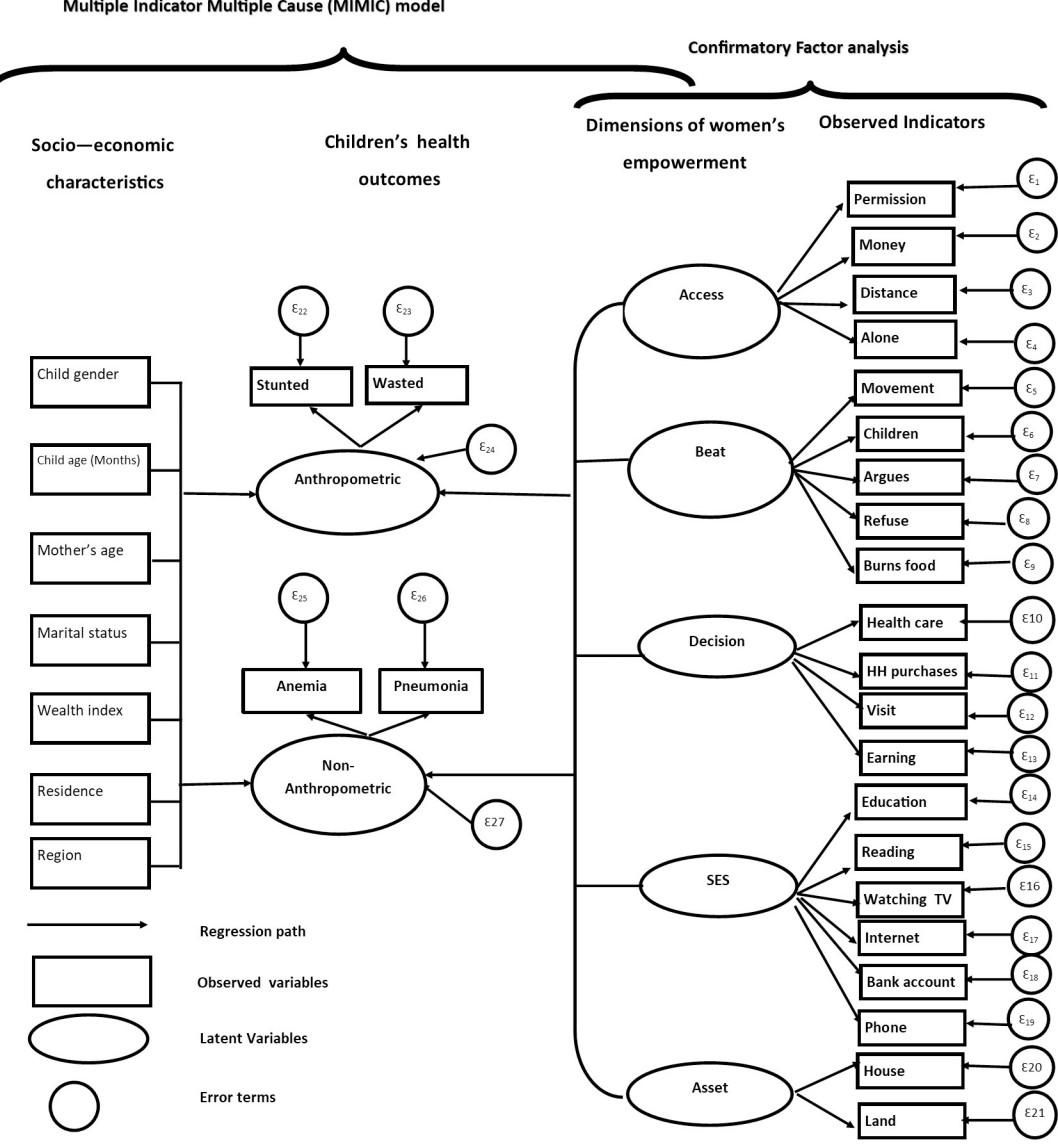

**Fig 1. Path diagram of the MIMIC model for women's empowerment and child health (Ethiopia).**

association of women's empowerment with the anthropometric causes of death (child malnutrition) with non-anthropometric causes of deaths for under-five children. In this study, since we have multiple indicators for both the women's empowerment and for health outcomes, we construct a latent variable for both.

On the right-hand side of Fig 1, Women empowerment is represented by five latent dimensions of empowerment derived from different observed indicators using exploratory and confirmatory factor analysis. Women's participation in decision making, barriers they face in access to health care, their attitudes towards wife-beating and education, exposure to media and asset ownership are considered as women empowerment dimensions. The observed indicators have been selected based on previous studies on women's empowerment and children's health in developing countries [25,32,52].

The path diagram in Fig 1 also shows that there is a direct influence from the women's empowerment and other control variables to the latent children's health status (the structural part of the MIMIC model). The exogenous variables that are assumed to determine child health include child gender and age and mother's characteristics (i.e., mother's age and marital status), household wealth index, place of residence and regional dummies. The selection of the variables was guided by studies on gender empowerment and health [16,19,25,26,53,54]. Mother's education was used as an indicator in constructing the women's socio-economic empowerment dimensions, hence we excluded it from the list of control variables to minimize multicollinearity.

## 4. Materials and methods

### 4.1 Data source and measurement

This study used data from the 2016 Ethiopian Demographic and Health Survey (EDHS). This survey is the fourth and most recent Demographic and Health Survey (DHS) conducted in Ethiopia following the 2000, 2005, and 2011 EDHS surveys [41]. It was conducted by the Central Statistical Agency (CSA) and Inner City Fund (ICF) International during January 18, 2016, to June 27, 2016 [41]. This sample for the 2016 EDHS was designed to provide estimates of key health indicators for the entire country, for both urban and rural areas, and the nine regions and the two administrative cities (See Fig 2).

Representative sample for the 2016 EDHS was chosen using two-stage stratified random sampling. In the first stage, each region was stratified into 21 urban and rural stratum and further into 17,185 urban and 67,730 rural Enumeration Areas (EAs) and of these, a sample of 202 urban and 443 rural EAs were randomly selected with probability proportional to EA size. Urban and rural EAs had an average size of 177 and 183 households, respectively [41]. In the second stage, a fixed number of 28 households were selected with an equal probability systematic selection from each EAs, resulting in a total sample household of 18,060 households. The final survey was conducted in 16,650 residential households, 5,232 in urban areas and 11,418

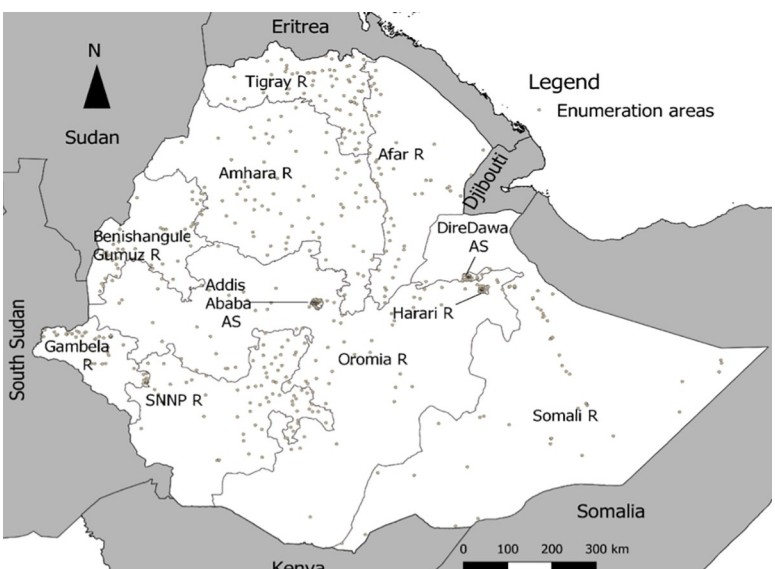

**Fig 2. Map of Ethiopia with regions and administrative states.** The enumeration areas have been randomly shifted, on average by 11 and 2 kilometers for rural and urban enumeration areas, respectively; R and AS stand for region and administrative state, respectively.

in rural areas. The survey was expected to generate an estimated 16,663 completed interviews with women age 15–49, 5,514 in urban areas and 11,149 in rural areas, and 14,195 completed interviews with men age 15–59, with 4,472 in urban areas and 9,723 in rural areas [41].

The data was collected through a survey questionnaire with five modules: the household, the woman, man, the biomarker, and the health facility. The Household module collected data on household and household member's characteristics. The Biomarker module collected information for each eligible household member (typically children under age 5, and women and men between age 15 and 49) on anthropometric measurements and levels of hemoglobin and records information about samples for biomarker testing. Women and men, aged between 15 and 49, and 15 and 59, respectively, were also interviewed using the Woman's and Man's modules, respectively. The Woman's module, in addition to questions about the woman, contains a birth history that is used to list all children (alive or dead) that the respondent has given birth to, along with the child's sex, date of birth, age, and survival status. The 2016 EDHS contains a total of 16,583 eligible women identified for individual interviews and interviews were completed with 15,683 women, yielding a response rate of 95% [41]. The birth history was the basis for selecting children under certain ages for the maternal health, immunization, child health, and nutrition sections of the questionnaire. The health facility module collected vaccination information for all children [41]. The sample is restricted to a sub-sample of 10,641 women between 15 to 49 years old and with at least one child under the age of five years in their care, since the focus of the paper is on child health outcomes. In this study, we considered not only women who are currently married and living with partner but also other categories of marital status such as widowed, divorced and separated. The variable marital status in our case is categorized as 1 if the women is married (either currently married or living with partner categories), 0 otherwise. In the DHS survey, women—either married or not married—responded to the questionnaire related to women's empowerment indicators and hence we use all women sample in our analysis.

## Ethics statement

The survey was implemented by the CSA of Ethiopia, which is mandated to collect all national data. The study protocol and data collection instruments were reviewed for adherence to ethical standards by the National Research Ethics Review Committee (NRERC). All study participants were asked for informed oral consent. The study uses publicly available secondary data without exposing any personal identification information.

**4.1.1 Health indicators.** Previous studies on child health in developing countries used a variety of indicators to represent a child's health outcome. Children's anthropometric indicators (Children's height-for-age, weight-for-age, and weight-for-height) are the most common measures of children's nutritional outcome [16,26,32,55,56]. Other child health indicators including child mortality, immunization, and treatment of diarrhea are also used as an outcome in some studies [20,25,28,42,43]. The current paper used both anthropometric indicators (child's stunting and wasting) and non-anthropometric indicators (child's exposure to Pneumonia and anemia status) as children's health indicators. Exposure of a children to diarrhea was not included in this study due to the high linear correlation with the exposure of a child to pneumonia. Detailed description of the health indicators is outlined below.

*Anthropometric indicators*. Studies suggest that anthropometric indicators are better measures of child health as they measure the nutritional status of infants and children using nutritional indices [55,56]. Child height/length, weight, and age data were used to calculate three indices: height-for-age, weight-for-height, and weight-for-age. Height-for-age measures whether a child is stunted or not. Stunting (low height-for-age) is a sign of chronic

undernutrition reflecting a lack of adequate nutrition over a prolonged period. Following the new WHO Child Growth Standards [57], we considered children whose height-for-age Z-score is below (i) minus two standard deviations (-2 SD) from the median of the reference population as short for their age (stunted), or chronically undernourished and (ii) below minus three standard deviations (-3 SD) as severely stunted. Weight-for-height index measures body mass in relation to body height describing current nutritional status. Following the new WHO Child Growth Standards [57], we considered children whose weight-for-height Z-score is (i) below minus two standard deviations (-2 SD) from the median of the reference population as thin (wasted), or acutely undernourished and (ii) minus three standard deviations (-3 SD) from the median of the reference population are considered severely wasted. Weight-for-age is a composite index of weight-for-height and height-for-age. Thus, weight-for-age, which includes both acute (wasting) and chronic (stunting) undernutrition, is an indicator of overall undernutrition. Children whose weight-for-age Z-score is below minus two standard deviations (-2 SD) from the median of the reference population are classified as underweight.

*Non-anthropometric indicators*. We used children's exposure to pneumonia and anemia status as non-anthropometric indicators. Pneumonia is a bacterial, viral, or fungal infection of one or both sides of the lungs that causes the air sacs, or alveoli, of the lungs to fill up with fluid or pus [58]. Acute Respiratory Infection (ARI) was considered as an indicator of child health that represents Pneumonia. EDHS 2016 collected information on under age 5 years children with symptoms of ARI in the 2 weeks preceding the survey. The questionnaire asks whether there was short, rapid breaths and the response categories were no, yes and do not know which were coded as one, zero and missing, respectively for further analysis.

Anemia, defined as a low blood hemoglobin concentration, usually results from poor nutrition, infection, or chronic disease. In the 2016 Ethiopia DHS, prevalence of anemia among children was collected with four categories: severe, moderate, mild, and not anemic. A dummy variable was created by collapsing the categories of severe, moderate and mild as anemic and the remaining as not anemic. We recoded "0" if severe, moderate and mild as anemic, "1" not anemic.

**4.1.2 Women's empowerment measures.** In this study, we operationalized women's empowerment using five dimensions: women's participation in decision making, attitudes towards wife-beating by husband, barriers faced by women in accessing health care and socio-economic empowerment. Women's participation in decision making was represented by four indicators (See Table 1). Women were asked "Who in your family usually has the final say on...?" on these four indicators. In the DHS, the responses were coded as "respondent", "husband or partner", "respondent and partner jointly", "someone else", "respondent and someone else jointly", and "decision not made/applicable" [41]. For this study, these response categories were further dichotomized to 1 if a woman had any decision (alone or jointly) and 0 if a woman had no say in one or more decisions. This is because women are considered to participate in household decisions if they make decisions alone or jointly with their husbands in all four household decisions [19,25,44,54].

We measure barriers faced by women in accessing health care for themselves by combining four indicators: getting permission to go to the doctor; getting money for advice or treatment; distance to a health facility; not wanting to go alone. Following [53], responses to these questions were dichotomized into 1 if the woman reported that the factor was not having a large problem, indicating a higher level of empowerment, and 0 if the woman reported that the factor was having a large problem, indicating a lower level of empowerment.

Attitudes towards wife-beating by husband includes five indicators, all related whether the woman agrees with husband's beating for wrong doings related to (a) burning food, (b)

**Table 1. Descriptive statistics (Ethiopian Demographic and Health Survey—2016).**

| Variable | Descriptions | Mean | SD |
|---|---|---|---|
| **Individual characteristics** | | | |
| Child's age (months) | Child's age in months | 29.16 | 17.61 |
| Child's gender | 1 if female; 0 otherwise | .48 | .49 |
| Mother's age (years) | Mother's current age in years | 29.55 | 6.61 |
| Marital status | 1 if married, 0 otherwise | .941 | .24 |
| Residence | 1 if rural;0 urban | .89 | .39 |
| **Women's empowerment variables** | | | |
| **Barriers to access to health care (Access)** | | | |
| Getting permission to go | 1 if "not a big problem"; 0 "big problem" | 0.62 | 0.49 |
| Getting money needed for treatment | 1 if "not a big problem"; 0 "big problem" | 0.38 | 0.48 |
| Distance to a health facility | 1 if "not a big problem"; 0 "big problem" | 0.399 | 0.49 |
| Not wanting to go alone | 1 if "not a big problem"; 0 "big problem" | 0.539 | 0.499 |
| **Beating justified (Beat)** | | | |
| Wife goes out without telling the husband | 1 if "no"; 0 "otherwise" | 0.51 | 0.50 |
| Wife neglects the children | 1 if "no"; 0 "otherwise" | 0.48 | 0.49 |
| Wife argues with husband | 1 if "no"; 0 "otherwise" | 0.52 | 0.49 |
| A wife refuses to have sex with husband | 1 if "no"; 0 "otherwise" | 0.58 | 0.49 |
| Wife burns the food | 1 if "no"; 0 "otherwise" | 0.55 | 0.49 |
| **Participation in household decision (Decision)** | | | |
| Health care | 1 if "respondent alone and jointly"; 0 "otherwise" | 0.79 | 0.41 |
| Large household purchases | 1 if "respondent alone and jointly"; 0 "otherwise" | 0.76 | 0.43 |
| Visits to family or relatives | 1 if "respondent alone and jointly"; 0 "otherwise" | 0.82 | 0.39 |
| Money husband earns | 1 if "respondent alone and jointly"; 0 "otherwise" | 0.75 | 0.44 |
| **Socioeconomic status (SES)** | | | |
| Highest educational level | 1 if secondary education and above; 0 otherwise | 0.04 | 0.26 |
| Frequency of reading newspaper | 1 if the women read a newspaper; 0 otherwise | 0.25 | 0.24 |
| Frequency of watching television and listening to a radio | 1 if the women listen to a radio or watching television; 0 otherwise | 0.47 | 0.47 |
| Frequency of using internet last month | 1 if the women use the internet; 0 otherwise | 0.11 | 0.11 |
| Has an account in a bank | 1 if the women have an account in a bank, 0 otherwise | 0.09 | 0.29 |
| Owns a mobile telephone | 1 if the women have a mobile phone; 0 otherwise | 0.16 | 0.37 |
| Employment | 1 if employed; 0 otherwise | 0.44 | 0.49 |
| **Asset ownership (Asset)** | | | |
| Owns a house alone or jointly | 1 if the women own a house alone or jointly; 0 otherwise | 0.68 | 0.47 |
| Owns land alone or jointly | 1 if the women own land alone or jointly; 0 otherwise | 0.55 | 0.49 |
| **Children's Health Outcomes** | | | |
| Pneumonia (Symptoms of ARI) | Dummy 1 if no, 0 if yes 2 weeks preceding the survey | .87 | .33 |
| Anemia level | 1 if not anemic, 0 otherwise | .42 | .49 |
| Weight-for-height (wasted) | Weight-for- height Z-score (WHZ) <-200 Standard Deviation | .10 | .30 |
| Height-for-age (Stunted) | Height-age-age Z-score (HAZ) <-200 Standard Deviation | .38 | .49 |

arguing with him, (c) going out without telling him, (d) neglecting the children, and (e) refusing to have sex with him [41]. Previously the responses for each indicator were "yes," "no" and "don't know." Following [44], in this study, the responses were further coded as "1" if the respondent says "no" and "0" if the respondent says "yes" which shows that no responses represent empowered women. We considered "don't know responses" as missing for each of the five indicators for attitudes towards wife-beating.

Socio-economic empowerment is measured using (i) education level, (ii) women's exposure to mass media, (iii) having an account in the bank, and (iv) owning a mobile telephone. A woman's exposure to media represent the frequency use of woman age 14–19 with access to media at least once a week. It consists of three indicators i.e. frequency of reading newspaper, frequency of watching television and listening to a radio and frequency of using internet. Since this variable shows the economic status of women in Ethiopia, we included them in the dimension of women's socio-economic variables [44]. In this case the woman is considered empowered if the she has access to these mass media indicators at least once in a week. The full list of indicators used in this study with their descriptions and summary statistics are presented in Table 1.

**4.1.3 Socio-demographic variables.** In addition to the empowerment dimensions, we include other children's, mothers' and household's characteristics to assess whether they are associated with child health. Children's characteristics include a child's age in months and gender. Mother's characteristics were represented with the mother's age in years and marital status. The household characteristics were represented by the wealth index of the household and residence (rural-urban), and regional dummies. Since the Demographic and Health Survey (DHS) did not collect directly household income, we used household wealth index as a measure of the household's socio-economic status. The wealth index is a composite measure of a household's cumulative living standard constructed using principal component analysis. The wealth index considered in this study is a categorical variable ranging from 1 to 5 in which 1 is the lowest and 5 is the highest. The categories are coded as "1" poorest, "2" poorer, "3" middle, "4" richer, and "5" richest.

Selection of socio-demographic variables included in this study was based on literature from previous studies on the association of women's empowerment and child health see e.g., [16,33,44]. (See also Table 1 for the description and summary statistics of the variables).

## 4.2 Methods

**4.2.1 Exploratory factor analysis and confirmatory factor analysis.** We use a combination of Exploratory Factor Analysis (EFA) and Confirmatory Factor Analysis (CFA) to construct the five latent empowerment dimensions. We use the full sample to perform both EFA and CFA. For this, we refer some studies and the studies indicated that implementing the EFA and CFA analyses using a full sample and split sample leads to similar results as far as methodological explanations can account for cases in which EFA and CFA lead to different conclusions based on the same sample (see e.g., [59]). These methodological issues include inadequate applications of EFA from previous literatures using the indicators, conservativeness of the CFA model and inappropriate applications of CFA [59]. We address the methodological issues in measuring the indicators related to women's empowerment as follows: first, based on previous literature we operationalized women's empowerment variables to check whether the indicators related to women's empowerment dimensions can be categorized in the same underlying latent factors. Second, some of the indicators related to women's empowerment, such as women's socio-economic status indicators, are not known in literature, we performed EFA at first. We then performed EFA by selecting all indicators related to women's empowerment to identify and determine whether a set of indicators stand together on one latent dimension and to establish a set of latent women's empowerment factors. Using the scree plot and Kaiser criterion (which suggests keeping factors with eigenvalues equal to or higher than 1) [60] to determine the optimal number of factors for further analysis and retained 5-factor scores (see S1 Fig). To measure factor correlation, we used Promax (Oblique) rotation assuming that the factors are highly correlated. Factor loadings $<|0.3|$ with lower

magnitude on any factor load with more than one factor were dropped. The five-factor solution, with factor loadings ranging from 0.5382 to 0.862, demonstrated good internal consistency ($\alpha$ = 0.85). All factor loadings are presented in the S1 Table.

The reliability of the model is evaluated by testing for internal consistency of a set of indicators using Cronbach's alpha. The coefficient of Cronbach's alpha ranges from 0 to 1 in which values of 0.9 or greater are considered excellent, 0.8 good, 0.7 adequate and 0.6 uncertain. Internal consistency is considered unacceptable for Cronbach's alpha value of less than 0.5 [61]. Our model had Cronbach's alpha coefficients ranging from 0.78 to 0.88 suggesting good reliability or internal consistency. Based on the identified factor loadings, the first factor related to wife beatings was labeled as "Beat." The second factor which includes women's highest-level education exposure to media and having a mobile phone and bank account is labeled as women's "Socio-economic status (SES)." The third factor is labeled as "Access" as it contains larger loadings of women's barriers to health care access indicators. Women's household "decision making" indicators is the fourth factor to describe women's empowerment. The final factor is "asset ownership" which contains two indicators namely women's house and land ownership (see S2 Table).

After identifying women's empowerment dimensions using EFA, CFA was applied to the same full dataset using the identified five factors and 21 indicators (see S2 Fig). We used CFA to verify the factor structure of the indicators related to women's empowerment and to test the hypothesis that a relationship between observed indicators and their underlying latent constructs exists. We allowed some of the indicator error terms to be correlated to better fit the CFA model. To do this we have calculated the modification indices of the CFA model. Factors which have higher modification indices were allowed their error terms to be correlated with each other. Model fit was assessed based on Root Mean Square Error of Approximation (RMSEA), Comparative Fit Index (CFI) and Tucker-Lewis Index (TLI). RMSEA and CFI are especially recommended for categorical data with large sample sizes. RMSEA values are interpreted as follows: RMSEA $\leq$0.05 good fit, $\leq$0.08 acceptable fit, and > 1.0 poor fit. Both CFI and TLI values of greater than 96% indicate a good fit [62]. Acceptable cut of value for SRMR is 0.08 or lower. The closer to 0, the better the fit of the model is [63]. A perfect correspond to Coefficient of Determination (CD) is 1. The closer the Value of CD to 1, the better the goodness of fit of the model. Literatures indicated that the acceptable cut off value of CD is depending on the number of exogenous latent variables [64,65]. Values of 0.67, 0.33 and 0.19 for endogenous latent variables in the inner path model are described as substantial, moderate and weak [64]. If endogenous latent variable explained by only a few exogenous latent variables, "Moderate" CD may be acceptable and if the endogenous latent variable relies on several exogenous latent variables, the CD value should exhibit at least a substantial level [65]. Hence, the overall goodness of fit of our model revealed the indicators were appropriately loaded onto their underlying factors, and the CFA model fits the data very well (RMSEA = 0.027; Close-fit test p-value = 1; CFI = 0.984; TLI = 0.980; and SRMR = 0.027, CD = 1) (see S2 Fig).

**4.2.2 Multiple Indicators Multiple Causes (MIMIC) model.** A Multiple Indicators Multiple Causes (MIMIC) model was employed to examine the relationship between women's empowerment and latent child health outcomes. The MIMIC model is composed of two components, namely the measurement model and a structural model [51,66]. The measurement model relates the observed health indicators (wasting, stunting, pneumonia, and anemia) to latent health while the structural model relates the latent health outcomes to observed exogenous variables (covariates).

The MIMIC model has been applied in different studies [16,19,50,52]. Define the latent child health, $\eta$, as a function of a set of observable exogenous variables $x_1, \ldots, x_n$, the general

form of the MIMIC model can be specified as:

$$y = \lambda\eta + \varepsilon \qquad (1)$$

$$\eta = \gamma'X + \zeta \qquad (2)$$

Where $y = (y_1,\ldots,y_n)$ is a vector of indicators of the latent variable $\eta$. $\gamma$ is the coefficient of $\eta$ and $X = (x_1,\ldots,x_n)$ is a vector of exogenous causes of $\eta$. Eq (1) represents the measurement model whereas Eq (2) is the structural part of the model. It is assumed that

$$E(\varepsilon\zeta') = 0,\ E(\varepsilon^2) = (\delta^2),\ \text{and}\ E(\varepsilon\varepsilon') = \Theta^2, \qquad (3)$$

with $\Theta$ being an $m\ \chi\ m$ diagonal matrix.

Since the latent child health outcome ($\eta$) is not observed, to estimate the coefficient of the model, we need to combine Eqs (1) and (2). The reduced-form of the model is then:

$$y = \lambda(\gamma'X + \zeta) + \varepsilon = \Pi'X + v \qquad (4)$$

Where the reduced form coefficient matrix is

$$\pi = \lambda\gamma' \qquad (5)$$

And the reduced-form disturbance vector is

$$v = \lambda\zeta + \varepsilon \qquad (6)$$

Since our health indicators are dichotomous responses, a conventional measurement component of the MIMIC model is specified as the multivariate normal distribution of 'latent responses' or 'underlying variables' [66,67]. The multivariate normality approach assumes that all variables in the model, exogenous and endogenous, are multi normally distributed [50]. In this case, the latent responses are linked to observed categorical responses via threshold models yielding probit measurement models [66]. Therefore, our MIMIC model was implemented using Generalized SEM with a probit link function to specify the categorical response variables in the model. Our MIMIC model contains two latent variables: anthropometric (constructed using wasting and stunting as indicators) and non-anthropometric (using anemia and pneumonia indicators). However, we excluded weight-for-age z-score as an indicator and from the estimation of the model because of the failure of the models to converge during the CFA and MIMIC model estimations. This is because weight-for-age Z-score is a composite index of weight-for-height and height-for-age.

The model was estimated using a maximum likelihood method with three different specifications for each latent health outcome. In the first specification, only the decomposed women's empowerment variables regressed on health outcome, in the second specification, we include children's and mother's characteristics together with the women's empowerment variable and in the third specification, we regressed only the composite women's empowerment index on child health. In all specifications, probability weights are applied to make the estimation results nationally representative in the case of Ethiopia. STATA version 14 [68] software was used to perform descriptive analyses as well as to build the structural equation models, including model building, path analyses, and regression estimations.

The relationship between the five dimensions of women's empowerment and children's health outcomes was described using box plot. We show the distribution of our data based on the five-number summary: minimum, first quartile, median, third quartile, and maximum median of each child health outcomes by the five dimensions of women's empowerment. Then, we conducted a median test to test the null hypothesis that the median empowerment

indices between the category of child health outcomes are identical. A median test is a special case of Pearson's chi-square test. We also calculate the marginal effects of women empowerment that are significantly associated with child health outcome. Our baseline specification includes women's empowerment variables, child's age and gender, mother's age, marital status, household wealth index, residence, and regional dummies.

## 5. Results

### 5.1 Descriptive result

Table 1 presents the sample characteristics for the variables included in the study. The average age of the child is 29 months and 48% of the children included in the study are female. The average age of the mothers is 29 years. 89% of the women live in rural areas. From the women's empowerment variables, 61% of the women reported that getting permission to go was not a big problem. However, 62% of the women reported that getting money for treatment was a big problem. Only 39% of women reported that distance to a health facility was not a big problem. More than half of the women reported that the husband was not justified in beating the wife while the wife goes out without telling the husband, wife argues with husband, wife refuses to have sex with her husband and when a wife burns the food. However, only 47% of the women reported a husband is not justified in beating the wife when the wife neglects the children. Regarding women's participation in household decision making, about 77% of the women participate in household decisions making alone or jointly with their partner in the four listed household decisions (own health care, large household purchases, visit their family and husband's earning.

The women show a disadvantage in terms of socio-economic status in which their educational level of shows that only 11% of women have secondary education and above. The level of exposure to media for women is exceptionally low in Ethiopia. The result shows only 24% of women read a newspaper at least once a week. Only 46% of the women were watching television and listening to a radio at least once a week. The majority of women also have low access to the internet. Only 10% of women have used the internet at least once a week. Only 9% of the women have a bank account and 16% of women have a mobile phone. Sixty-eight percent and 55% of women own a house and land alone or jointly. This percentage is higher as compared to other socio-economic indicators.

Fig 3 displays the difference in distribution of the five dimensions of women's empowerment categorized by the four children's health outcomes (i.e., stunted (panel A), wasted (panel B), anemia (panel C) and pneumonia level (panel D)). The women's empowerment indices are presented on the vertical axis and children's health outcomes on the horizontal axis. The middle line in the box shows the median, the bottom and top of the box show the 25th & 75th quartiles, respectively. So the height of the box is the inter-quartile range (IQR). The figure shows that the median value of all women empowerment components for non-stunted children is higher than the stunted children. Similarly, mothers of non-wasted children have a higher level of empowerment in all dimensions except for asset ownership. The figure indicates that women's empowerment dimensions are positively related with the health of children. We observed the same pattern for anemia and pneumonia status of the children. However, the median index of women's empowerment dimension represented by 'beat' is higher for anemic children and the median of women's empowerment index with regard to socio-economic status and asset ownership is higher for children's pneumonia level which represent by "yes" (See Pane C and D of Fig 3).

We conducted nonparametric equality-of-medians test to check the null hypothesis that the median empowerment indices between the category of child health outcomes are identical.

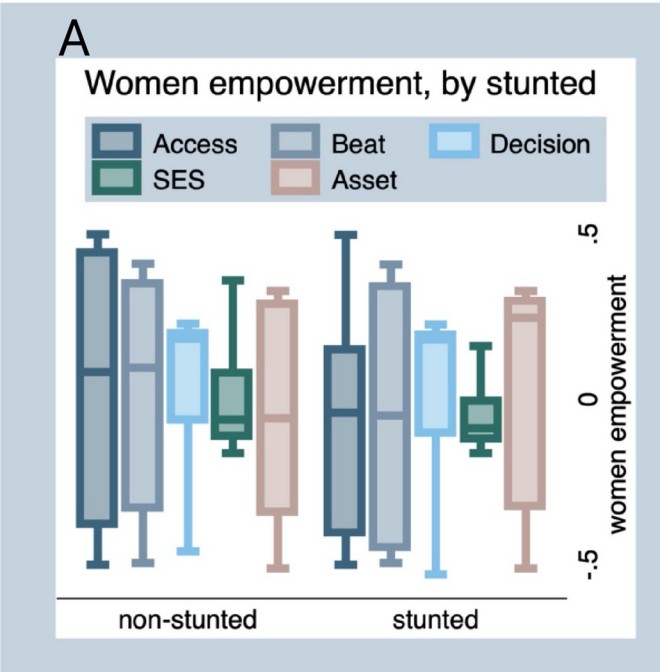

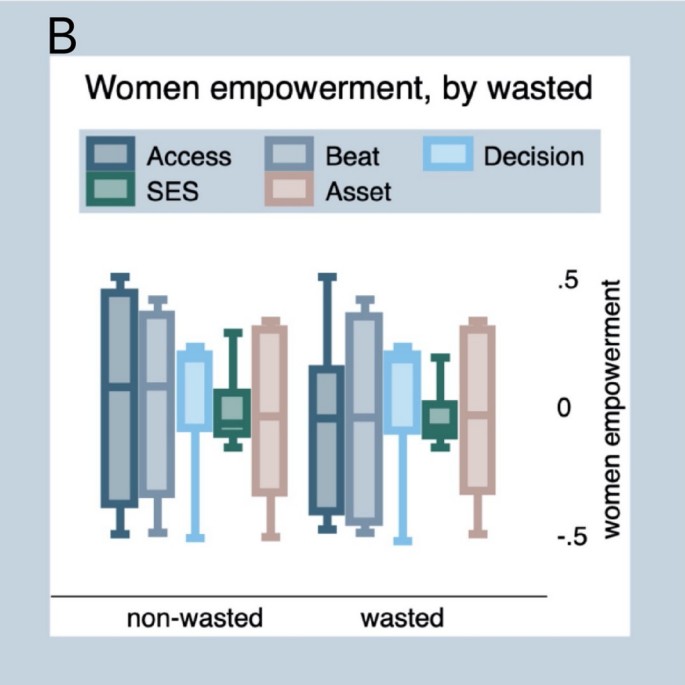

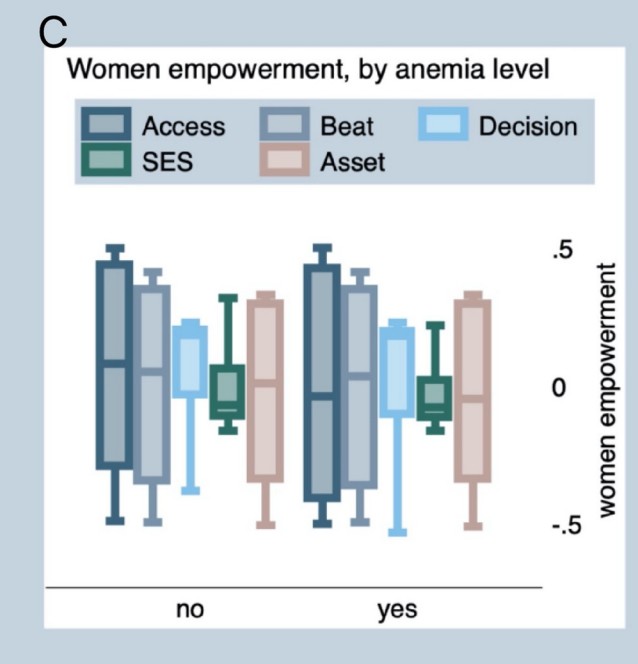

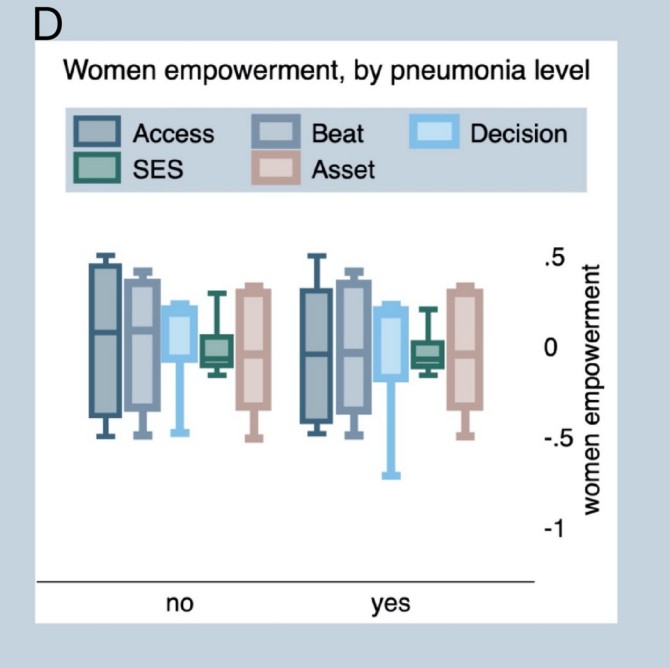

**Fig 3. The difference in distribution of the five dimensions of women's empowerment categorized by child health outcomes.**

The test result indicates that the medians of women empowerment components are statistically different between stunted and non-stunted children as well as between wasted and non-wasted children (Pr = 0.000), except, for asset ownership, for which we do not observe a significant difference between wasted and non-wasted children (Pr = 0.718). We observe significant

**Table 2. Regression results of the measurement part of the MIMIC model.**

| Health Indicators | Model-1 | Model-2 | Model-3 |
|---|---|---|---|
| *Anthropometric Health* | | | |
| Height-for-age (stunted) | 1 (.) | 1 (.) | 1 (.) |
| Constant | -0.330*** (0.026) | -0.323* (0.184) | -0.304* (0.184) |
| Weight-for-height (wasted) | 0.497** (0.230) | -0.353** (0.156) | -0.346** (0.159) |
| Constant | -1.295*** (0.029) | -1.319*** (0.074) | -1.325*** (0.072) |
| *Non-Anthropometric Health* | | | |
| Pneumonia | 1 (.) | 1 (.) | 1 (.) |
| Constant | -1.197*** (0.039) | -0.983*** (0.072) | -0.995*** (0.072) |
| Anemia | 1.559*** (0.585) | 4.783*** (1.391) | 5.341*** (1.853) |
| Constant | 0.184*** (0.037) | 1.158*** (0.211) | 1.221*** (0.255) |

Standard errors in parentheses

* $p < 0.10$

** $p < 0.05$

*** $p < 0.01$.

difference for all empowerment dimensions between anemic and non-anemic children and between children exposed and not exposed to pneumonia, except for beating dimension which was insignificant between anemic and non-anemic children (Pr = 0.413) and for SES (Pr = 0.933) and asset ownership (Pr = 0.997), in which there were no statistically significant difference between children exposed and not exposed to pneumonia. Details of the median test results is provided in the S2 Table.

## 5.2 Measurement part of the MIMIC model

Table 2 presents the unstandardized result of the measurement part of the model. This measurement component shows the association between the latent health status variables and its observed indicators. For identification purposes, the factor loadings for height-for-age and pneumonia were set to one to form the scale of the latent variable [16]. The estimated coefficients for weight-for-height and anemia indicators are statistically significant confirming the existence of 'single' underlying latent variable and that this latent variable is represented by each of the health indicators. The positive signs on the weight-for-height z-score variables imply that a household with a higher number of wasted children is associated with lower overall children's health status.

## 5.3 Structural part of the MIMIC model

Table 3 reports the results of the structural part of MIMIC model. The models were constructed using three specifications for both the anthropometric and non-anthropometric children's health outcomes as described in the methods section. Results for the anthropometric outcomes show that the associations between women's empowerment dimensions and children's health outcomes are not statistically significant, with the exception of the women's socioeconomic dimensions. This result is robust even after controlling for individual-level characteristics. The results show that higher empowerment of women in terms of their socioeconomic status is associated with less likelihood of their children being stunted or wasted -0.78 ($p < 0.05$). The overall empowerment index (in model 3 of the anthropometric indicator) is also found to be statistically significant for the anthropometric health indicators -0.49 ($p < 0.05$). Child's age is positively associated with the status of being stunted or wasted. This

**Table 3. Results of the structural component of the MIMIC model.**

| Variables | Anthropometric Health Indicator | | | Non-Anthropometric Health Indicator | | |
|---|---|---|---|---|---|---|
| | Model-1 | Model-2 | Model-3 | Model-1 | Model-2 | Model-3 |
| Access to healthcare | 0.0471 (0.067) | 0.031 (0.064) | - | -0.1695** (0.075) | 0.001 (0.017) | - |
| Beating | 0.009 (0.064) | 0.048 (0.076) | - | 0.042 (0.048) | 0.022 (0.017) | - |
| Decision | -0.041 (0.066) | -0.0205 (0.082) | - | -0.169*** (0.059) | -0.059** (0.024) | - |
| SES | -1.189*** (0.228) | -0.776** (0.312) | - | -0.324*** (0.122) | -0.065 (0.055) | - |
| Asset ownership | 0.016 (0.073) | -0.133 (0.092) | - | -0.115** (0.056) | 0.005 (0.019) | - |
| Overall empowerment | - | - | -0.487** (0.319) | - | - | -0.067 (0.063) |
| Child's age (months) | - | 0.016*** (0.002) | 0.016*** (0.002) | - | -0.005*** (0.001) | -0.005*** (0.002) |
| child's gender (female) | - | -0.137** (0.054) | -0.139** (0.055) | - | -0.005 (0.009) | -0.006 (0.009) |
| Mother's age (years) | - | -0.007** (0.004) | -0.008** (0.004) | - | -0.001 (0.001) | -0.001 (0.001) |
| Married | - | -0.048 (0.109) | -0.070 (0.111) | - | 0.018 (0.021) | 0.016 (0.020) |
| *Household wealth index* | - | | | - | | |
| Poorer | - | -0.092 (0.082) | -0.093 (0.084) | - | 0.039** (0.016) | 0.038** (0.016) |
| Middle | - | -0.236** (0.092) | -0.239** (0.094) | - | -0.061*** (0.021) | -0.060*** (0.022) |
| Richer | - | -0.242** (0.100) | -0.254** (0.104) | - | -0.048** (0.019) | -0.049** (0.019) |
| Richest | - | -0.319** (0.124) | -0.359*** (0.130) | - | -0.087*** (0.030) | -0.088*** (0.032) |
| Rural | - | 0.045 (0.122) | 0.062 (0.125) | - | -0.006 (0.029) | -0.0052 (0.027) |
| Region dummy | - | yes | yes | - | yes | yes |
| N | 9005 | 9005 | 9005 | 9919 | 9919 | 9919 |

Standard errors in parentheses

* p<0.10

** p<0.05

*** p<0.01; the number of observations are lower than 10641 and the number of observations differ across observation are due to missing values in outcome variables of interest.

implies that older children are more likely to be related with the status of being stunted or wasted, which may be due to their higher nutritional requirement as they age. Female children are less likely to be stunted or wasted as compared to their male counterparts. Being in the middle, the richer and richest quintile of the household wealth index are also associated negatively with being stunted or wasted implying that household wealth has a positive impact on children's wellbeing. Children of older mothers are also less likely to be stunted.

The results from the structural component of the MIMIC model for the latent non-anthropometric health outcomes (anemia and pneumonia) show that the dimensions of barriers to access to health care, decision making, socioeconomic status, and asset ownership were negatively and statistically significantly associated with pneumonia and anemia indicator of latent health before controlling individual socio-economic characteristics. However, only the decision-making dimension is found to be statistically significant after controlling for both children's and mothers' characteristics and regional dummies (Model-2) -0.059 (p<0.05). Results from the third model show that the composite women's empowerment index is not statistically significant for non-anthropometric health indicators. Children's age and household wealth index remained to be statistically significant. The negative sign of the coefficient child age shows that older children are less likely to suffer from anemia and pneumonia. Similarly, wealth index is positively and significantly associated negatively with anemia and pneumonia indicating its positive impact on children's health.

Table 4 presents the marginal effects of women's empowerment that are significantly associated with children's health outcomes. The table shows the coefficient and standard error for

**Table 4. The marginal effects of women's empowerment on children's health status.**

| Women's empowerment Dimensions | Child health indicators | | | |
|---|---|---|---|---|
| | Stunted | Wasted | Pneumonia | Anemia |
| Socio-economic Status (SES) | -.290*** (.088) | -.032** (.0505) | - | - |
| Household decision making (Decision) | - | - | -.025*** (.008) | -.108***(.026) |
| Overall empowerment (WEI) | -.164** (.089) | -.1380*** (.0529) | - | - |

Standard errors in parentheses, * p<0.10

** p<0.05

*** p<0.01.

the marginal effect of the decomposed empowerment dimensions as well as the composite women's empowerment index that are significantly associated with children's health status. The results show that women's socio-economic empowerment dimension is associated with a 26.3% (p<0.001) (prediction in the probability of being stunted (height-for-age Z-score) and a 3.24% (p<0.05) reduction in the probability of being wasted (weight-for-height Z-score). This shows that women's empowerment regarding their socio-economic status has a higher association on stunting than wasting. This result is in congruence with previous studies on the effect of socioeconomic status on stunting and wasting [32,69]. For non-anthropometric dimensions, the result shows that higher empowerment with regard to women's household decision making is associated with a 2.5% (p<0.001) reduction in the probability of having pneumonia and a 10% (p<0.001) decrease in the probability of being anemic for the children. This may imply that women's higher decision-making power has a significant effect on children's health status. Since women's decision making was not significant for the child's health status indicated by being wasted and/or stunted, their marginal effect is not calculated. Based on the marginal effects of the composite women's empowerment index, a one unit increase in the women's empowerment is associated in a reduction by16.4% probability of being stunted and a reduction by 13.80% probability of being wasted.

After the estimation of the Generalized SEM MIMIC model, we evaluated the goodness of fit of the various model specifications using the Akaike Information Criterion (AIC) and Bayesian Information Criteria (BIC). Since other goodness of fit indices are not available under Generalized SEM responses, we use these fit indices to compare the different specifications of MIMIC models. Smaller AIC and BIC values indicate a better fit model. In this case, model 2 for both anthropometric and non-anthropometric health indicators show a good fit model as compared to the other model specification (See Table 5) below.

## 6. Discussion

### 6.1 Women's empowerment and anthropometric health status

Our finding reveals that the association of women's empowerment and children's health outcome varies depending on the specific dimension of women's empowerment and the health

**Table 5. The goodness of fit of the models.**

| Variables | Anthropometric Health Indicator | | | Non-Anthropometric Health Indicator | | |
|---|---|---|---|---|---|---|
| | Model-1 | Model-2 | Model-3 | Model-1 | Model-2 | Model-3 |
| Observations | 9,005 | 9,005 | 9,005 | 9,919 | 9,919 | 9,919 |
| Log pseudolikelihood | -9,434.42 | -9,163.26 | -9,171.94 | -9,569.92 | -9,050.57 | -9,069.19 |
| AIC | 18,884.84 | 18,382.53 | 18,391.89 | 19,157.83 | 18,155.14 | 18,184.38 |
| BIC | 18,941.68 | 18,581.48 | 18,562.42 | 19,222.65 | 18,349.6 | 18,350.03 |

outcome variable under consideration. We find that only one component of women empowerment (i.e., women's socio-economic status that captures women's education, exposure to media, having a bank account and mobile phone) was statistically significant and positively associated with anthropometric health indicators of children after controlling for individual-level characteristics. This is in line with the thesis that more educated mothers and mothers who have more access to media, have access to health and nutrition-related information for their children, which in turn improves children's health and their nutritional status [28,56]. Hence, women's socioeconomic status in a household is vital for children's health and empowering women in terms of their socioeconomic status may improve their children's anthropometric health status. Some previous studies on gender empowerment and child health also acknowledge the importance of socio-economic empowerment of women on anthropometric child health outcomes [28,56,70].

Other empowerment dimensions (i.e., access to healthcare, decision-making power, attitudes towards wife-beating by husband, and asset ownership) are insignificant, which suggests that the empowerment of women in arenas other than socioeconomic status may not help improving children's health status of stunting and wasting. This finding is consistent with the findings of some previous studies that reported no statistically significant association between some women's empowerment dimensions (e.g., decision making, access to healthcare, beating or domestic violence) and children's nutritional status of height-for-age and weight-for-height [16,26,29,55,71]. For example, [16] found no significant association between women's empowerment in agriculture and child health status in Northern Ghana. Using cross-sectional data from the 2013–14 DHS data, in the Democratic Republic of Congo [71], found women's decision making power were not significantly associated with stunting or wasting in children. Similarly, [32] found that women's decision making authority has a weak effect on children's health in sub-Saharan Africa. [26] made a similar conclusion in the case of Mozambique. A study by [55] also found the mother's decision-making power and freedom of mobility were not statistically and significantly associated with child health outcomes, particularly stunting. However, our overall women's empowerment index, encompassing all components significantly associated with children's nutritional status of stunting and wasting revealed focusing on full-fledged empowerment of women would help to improve child health outcomes in relation to stunting and wasting.

## 6.2 Women's empowerment and non-anthropometric health status

Our result suggests that women's empowerment in terms of household decision making dimension (composed of women's status about decision making to their healthcare, large household purchases, visit family and friends and the money husband earn) related to children's non-anthropometric health indicators. This relationship is robust when we control for household, child and mother characteristics. Even though other empowerment dimensions (namely, access to healthcare, socioeconomic status, and asset ownership) are found to be related at first, their significance disappeared when we control for the other variables. These findings suggest that only empowerment concerning decision making has a robust effect in improving child non-anthropometric health indicators meaning that the children are less likely to have pneumonia and anemia when the mother has greater decision-making power. A recent study by [22], confirms consistent associations between women's empowerment in household decision making and indicators of child non-anthropometric health status. According to their finding, "women's instrumental agency," which they measured as participation in household decision making is found more relevant for anemia indicator. One possible explanation regarding the association of women's household decision making and non-

anthropometric child health indicator is that women's household decision making power is women's day to day activity in the household, which increases access to child health services and increases their freedom of movement [72]. Also, if the woman involve in making decisions on their partner's earnings, they may allocate their money for their children.

## 6.3 Women and household characteristics, anthropometric and non-anthropometric child health outcomes

Child age is positively associated with being stunted or wasted indicating that as children's age increases, they are more likely to be stunted or wasted. Despite this finding is not consistent with some previous studies that found a negative association between a child's age and being stunted or wasted [16], it suggests accumulation of the health problems over time and the higher nutritional requirement with age. Female children are less likely to be stunted or wasted as compared to male counterparts. A mother's age is negatively related to children's being stunted or wasted. Being in the middle, the richer and richest quintile of the household wealth index are also associated negatively with being stunted or wasted meaning that household wealth has a positive impact on children's wellbeing. Children of older mothers are also less likely to be stunted.

Concerning non-anthropometric child health outcomes, child age and household wealth index are found to be statistically significant and negatively associated with child health indicators of anemia and pneumonia. The negative sign of the coefficient child age shows that children are less likely to have anemia and pneumonia when their age increases and children from wealthy households are less likely to be exposed to both health problems. Similarly, wealth index is associated negatively with anemia and pneumonia meaning that household wealth has a positive impact on children's health.

## 7. Conclusions

Using data from 2016 Ethiopian Demographic and Health Survey (EDHS), this study empirically investigates the association between women's empowerment and different child health outcomes: stunting, wasting, anemia, and pneumonia. We considered a multidimensional women's empowerment represented by barriers faced by women in accessing healthcare, household decision making, attitudes towards wife-beating by husband, socio-economic status and asset ownership. Two latent health indices are constructed using two indicators of the nutritional status of a child (stunted or wasted) and two non-nutritional child health indicators (pneumonia and anemia). The data is analyzed using the Multiple Indicator Multiple Causes (MIMIC) model. Our findings indicated in Ethiopia women's socio-economic empowerment and household decision making are important for improving their health status. Hence, gender-specific policies focusing on increasing women's access to education, employment, media, and information, and promoting saving and their participation in household decision making are some of the strategies for improving their children's health and wellbeing. Findings contribute to efforts by policymakers to promote women empowerment and its associated effects on health promotion.

Our study has two limitations. First, our empowerment measures are limited to DHS data and we do not explore more empowerment dimensions expect for empowerment dimensions only available in the DHS survey. Second, the cross-sectional nature of the DHS survey limits our analysis to a single period. By considering the multidimensionality and dynamic nature of empowerment, future studies may benefit by collecting primary data to a specific context to incorporate more dimensions of women's empowerment (e.g., psychological, political, legal aspects) and to understand the change in empowerment and health over time.

## Supporting information

**S1 Fig. Scree plot of eigenvalues after factor analysis.**
(DOCX)

**S2 Fig. Confirmatory factor analysis model for women's empowerment (EDHS-2016).**
(DOCX)

**S1 Table. Result of EFA.**
(DOCX)

**S2 Table. The difference in distribution of the five dimensions of women's empowerment by the four child health outcomes.**
(DOCX)

## Author Contributions

**Conceptualization:** Solomon Kibret Abreha, Solomon Zena Walelign.

**Data curation:** Solomon Kibret Abreha.

**Formal analysis:** Solomon Kibret Abreha.

**Investigation:** Solomon Kibret Abreha.

**Methodology:** Solomon Kibret Abreha, Yacob Abrehe Zereyesus.

**Resources:** Solomon Kibret Abreha.

**Supervision:** Solomon Zena Walelign, Yacob Abrehe Zereyesus.

**Validation:** Solomon Zena Walelign, Yacob Abrehe Zereyesus.

**Visualization:** Solomon Zena Walelign, Yacob Abrehe Zereyesus.

**Writing – original draft:** Solomon Kibret Abreha.

**Writing – review & editing:** Solomon Kibret Abreha, Solomon Zena Walelign, Yacob Abrehe Zereyesus.

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
