## [Decision Letter · Decision Letter 0]

10 Feb 2020

PONE-D-20-00349

Associations between Women’s Empowerment and Child Health Status in Ethiopia

PLOS ONE

Dear Dr Abreha,

Thank you for submitting your manuscript to PLOS ONE. After careful consideration, we feel that it has merit but does not fully meet PLOS ONE’s publication criteria as it currently stands. Therefore, we invite you to submit a revised version of the manuscript that addresses the points raised during the review process.

The reviewers had substantial comments about the paper, both in terms of the methods but also the presentation of the data. Please review and address each of their comments. I would also suggest that you have someone edit the English grammar. 

We would appreciate receiving your revised manuscript by March 10, 2020. To enhance the reproducibility of your results, we recommend that if applicable you deposit your laboratory protocols in protocols.io, where a protocol can be assigned its own identifier (DOI) such that it can be cited independently in the future. For instructions see: http://journals.plos.org/plosone/s/submission-guidelines#loc-laboratory-protocols

We look forward to receiving your revised manuscript.

Kind regards,

Mellissa H Withers, PhD, MHS

Academic Editor

PLOS ONE

Journal Requirements:

1. Please provide the relevant ethical documentation accompanying the 2016 DHS survey. Please clarify whether data were analyzed anonymously. If data were not analyzed anonymously please include the consent procedure that accompanied the 2016 DHS survey and confirm your study falls within what particpants have consented to.

Reviewers' comments:

Reviewer's Responses to Questions

**Comments to the Author**

1. Is the manuscript technically sound, and do the data support the conclusions?

Reviewer #1: Yes

Reviewer #2: Partly

2. Has the statistical analysis been performed appropriately and rigorously? 

Reviewer #1: Yes

Reviewer #2: I Don't Know

3. Have the authors made all data underlying the findings in their manuscript fully available?

Reviewer #1: Yes

Reviewer #2: Yes

4. Is the manuscript presented in an intelligible fashion and written in standard English?

Reviewer #1: Yes

Reviewer #2: No

5. Review Comments to the Author

Reviewer #1: I was very pleased to be invited to review this paper. The authors demonstrate a fairly good understanding of the literature regarding women’s empowerment and child outcomes, though they may be overstating that it is limited, as many of the existing articles are not cited. They could possible specify that it is limited in this particular context. They have also approached their impressive analytic intentions with the appropriate level of data, namely DHS. The extensive and thorough measures of female empowerment is the contribution of the paper but the introduction, literature review, and conclusion seriously limit the implications of this paper. There needs to be a deeper dive into the existing literature, the findings of this paper, and the implications for a variety of stakeholders.

Section 1

The second paragraph about rates could be more succinctly presented.

The authors do a good job of explaining the gap that exists, they just need to add more of the existing literature to the review in order to convincingly say that this gap exists across a majority of papers on this topic.

Section 2

The authors are thorough in their discussion of Ethiopia. While context is important, this should be presented more succinctly and replaced with information about the existing studies on women’s empowerment and child health. If this paper is indeed addressing the gap in this pretty expansive literature, I need to be convinced that this gap exists. Section 3 hints at this in the first paragraph, but it does not adequately situation this paper in the existing literature.

Section 4.1.2

It is helpful to present alpha scores

What is the justification for the all or nothing coding of the decision-making variable?

It is not clear if the barriers to healthcare are combined into one variable.

There are two paragraphs that explain the attitudes toward wife-beating

It would be helpful to justify (citations) why exposure to media is a measure of socio-economic empowerment

Section 5.2

Numbers should be used in the results opposed to just positively or negatively associated

Section 6.1

There was no mention of a bank account or mobile phone in the measures section

Section 7

A main contribution of this paper is the thorough investigation of different avenues of women’s empowerment. However, the literature review and the conclusion does not convince me that they are covering all important measures of empowerment or how those different measures would/do work differently to influence child health. Why could certain empowerment measures matter over others for child’s health? The conclusion is seriously lacking considering the numerous findings, which are the actually, most measures of empowerment don’t statistically seem to matter. This could be a troubling take away, so explaining it is of utter importance.

Typographical and grammar edits- this paper needs a thorough read through, particularly the discussion and conclusion sections. A few are mentioned below, but there is a need for greater clarity at the sentence level.

pg 2 line 28 “Even those that with an…”

pg 3 line 59 “Reducing THE child mortality rate’

pg 3 line 70 “helps improving”

pg 7 line 143 “with a greater agency”

Pg 25 lines 522-525 do not use consistent citations

Pg 26 lines 552-556 are not clear

Pg 28 line 583 “giving the…”

Reviewer #2: Overall

The paper uses the latest Ethiopian DHS to examine the association between women’s empowerment and child health status. The authors use SEM to fit a measurement and a structural component and to simultaneously examine how women’s empowerment is associated with both anthropometric and non-anthropometric child health outcomes. The findings contribute to the evidence base on how women’s empowerment is associated with different child health outcomes, and can help inform local interventions to help improve women’s empowerment and child health.

Major concerns

Measurement model – it is unclear why the authors conceptualize and fit two separate anthropometric and non-anthropometric latent health variables. It is also not clear why these constructs are conceptualized as latent construct rather than as observed variables which is more common in the literature on women’s empowerment and children’s health status. Additionally, why was weight-for-age Z-score not included in the anthropometric latent construct?

Several details are not included in the methods section, which would make replication of the results difficult. Was half the sample used for EFA? If so, how was this sample selected? How was the cut-off of 0.3 for keeping factor loadings chosen? It would be helpful to include all factor loadings in supplemental table 3, and then indicate (in bold, italic or some symbol) which factor loadings made the cut-off. It is informative to know how close to the cut-off factor loadings were.

With respect to CFA, supplemental figure 2 shows that indicator error terms were allowed to be correlated. It is not clear from the methods section why this is the case; if this was an a priori hypothesis or if the authors tested multiple CFA models allowing/not allowing error terms to be correlated. What estimator was used to fit the CFA model? Further, the authors provide no evidence of discriminant validity, e.g. what are the correlations between factors.

Please indicate what software was used for data analysis.

How was missing data handled in the EFA, CFA and MIMICs models?

Line 407, specify in the methods section what test was used to test for differences in empowerment indices and how statistical significance was defined. It may be helpful to create a table with all the results rather than to selectively present some of them in text.

Table 2, the methods section should describe how the marginal effects of women’s empowerment on child’s health were calculated. The table is somewhat confusing with empowerment dimensions as rows but model 2 including the empowerment index. The results presented after Table 3 indicate that model 2 was the best fit model, but marginal effects are not calculated for this model. How come?

The authors should consider moving the measurement results and Table 3 before the MIMICs results, since the measurement results inform the outcomes which are then used in the MIMICs model. The health indicators listed in Table 3 are not in line with the definitions in the text. The authors should be more careful and consistent in how they refer to the health indicators throughout to improve the clarity of the article.

The use of causal language is misleading. Given the cross-sectional nature of the data and the analytic methods employed, the authors can only assess the association between women’s empowerment and child health status. Throughout the manuscript, the authors should clean up the causal language (e.g., “impact”, “affect”, “effect”) and replace it with the appropriate non-casual language.

The discussion on child, women and household characteristics associated with anthropometric and non-anthropometric health indicators can be cut down. These are largely known from the literature. The discussion will benefit from a more in-depth discussion on why certain domains of women’s empowerment are associated with child outcomes while others are not, and how these findings are, or are not, specific to Ethiopia. Lines 542-545 are a bit unclear with respect to the hypotheses the authors are making. The authors argue that “if the woman decides on their partner’s earnings, they may allocate their money for their children”. Yet the indicator for decision-making with respect to partner’s earnings was defined as 1=“respondent alone or jointly”, which does not necessarily substantiate the argument made by the authors. It might not be women’s ability to decide on earnings, but rather their involvement in the decision-making process. The discussion should be a bit more nuanced with respect to the interpretations and implications given the data limitations acknowledged by the authors.

Minor concerns

Overall – the manuscript can benefit from copyediting; there are typos, unclear sentences, etc. Further, the authors should try to use more consistent language to improve the clarity of the paper. For example, switching between nutrition indicators and anthropometric indicators makes it unclear whether these are two distinct types of indicators. Please spell out all abbreviations the first time they are used, e.g. ARI line 263.

Abstract – unclear whether “empowered women” refers to a binary variable where women were considered empowered or not empowered, or a continuous variable where higher empowerment was associated with lower likelihood of stunting.

Introduction – the introduction should be strengthened to provide relevant statistics on the prevalence of wasting, stunting, anemia and pneumonia as well as on the state of women’s empowerment. The local context is heavily described in terms of infant and child mortality, which are not outcomes the paper focuses on. Further, it would be helpful to provide more context on what previous studies, if any. in Ethiopia (neighbouring countries or Sub-Saharan Africa) have found on the association between women’s empowerment and child health outcomes, and what are the specific gaps in the literature that the paper is trying to address.

Conceptual framework – pleases provide more detail in the text or the caption for Figure 1 on what is not depicted, e.g. variances for exogenous variables, etc. The conceptual framework can be more explicitly about why the authors expect women’s empowerment to be associated with child outcomes. How were the domains of women’s empowerment chosen?

Indicators

Line 226, were only married women and women living with partner kept in the analysis? Several of the DHS indicators used in the measurement model are only collected for married women/women living with their partner.

Indicate the cut-off used to define anemia

Women’s empowerment indicators – were “don’t know” responses coded as missing?

The paragraph on attitudes towards wife beating was repeated, i.e. para lines 288-293, and para lines 294-300.

Please be specific throughout that household wealth quintile was included as a categorical variable. It is not always clear whether the continuous wealth index was used or the wealth quintile.

Tables and figures - Please spell out all abbreviations in the main and supplemental tables and figures

6. PLOS authors have the option to publish the peer review history of their article (what does this mean?). If published, this will include your full peer review and any attached files.

Reviewer #1: No

Reviewer #2: No

---

## [Author Response · Author response to Decision Letter 0]

14 Apr 2020

All the response to the reviewers have been uploaded separately in the attachment. Kindly check the attached response to reviewers file. Thank you

---

## [Decision Letter · Decision Letter 1]

18 May 2020

PONE-D-20-00349R1

Associations between Women’s Empowerment and Children's Health Status in Ethiopia

PLOS ONE

Dear Mr. Abreha,

Thank you for submitting your manuscript to PLOS ONE. After careful consideration, we feel that it has merit but does not fully meet PLOS ONE’s publication criteria as it currently stands. Therefore, we invite you to submit a revised version of the manuscript that addresses the points raised during the review process.

We would appreciate receiving your revised manuscript by Jul 02 2020 11:59PM. To enhance the reproducibility of your results, we recommend that if applicable you deposit your laboratory protocols in protocols.io, where a protocol can be assigned its own identifier (DOI) such that it can be cited independently in the future. For instructions see: http://journals.plos.org/plosone/s/submission-guidelines#loc-laboratory-protocols

We look forward to receiving your revised manuscript.

Kind regards,

Mellissa H Withers, PhD, MHS

Academic Editor

PLOS ONE

Reviewers' comments:

Reviewer's Responses to Questions

**Comments to the Author**

1. If the authors have adequately addressed your comments raised in a previous round of review and you feel that this manuscript is now acceptable for publication, you may indicate that here to bypass the “Comments to the Author” section, enter your conflict of interest statement in the “Confidential to Editor” section, and submit your "Accept" recommendation.

Reviewer #2: All comments have been addressed

2. Is the manuscript technically sound, and do the data support the conclusions?

Reviewer #2: Yes

3. Has the statistical analysis been performed appropriately and rigorously? 

Reviewer #2: Yes

4. Have the authors made all data underlying the findings in their manuscript fully available?

Reviewer #2: Yes

5. Is the manuscript presented in an intelligible fashion and written in standard English?

Reviewer #2: Yes

6. Review Comments to the Author

Reviewer #2: Thank you for the opportunity to review this paper. The revised manuscript is much improved. The authors have addressed the majority of my concerns in the text or through their responses. A few outstanding comments and suggestions:

1. EFA/CFA - conducting EFA and CFA on the full sample is a bit unorthodox. Typically, one random split half is used for EFA, and the remaining half for CFA. It would be helpful if the authors provide further rationale or a reference for using the full sample for these analyses.

2. Supplemental table 1 - I suggest you move this table to the main text. Rereading the paper now, I find myself wanting to see a description of the sample.

3. Introduction - thank you for including the additional literature. This is very helpful in putting your study in context. Lines 99-102, I agree that this is a limitation of the previous literature, but some the studies never aimed to assess infectious disease or causes of death, e.g. Heckert et al. aimed to asses only nutritional status. I suggest the authors re-phrase this sentence to more properly cite the literature. Lines 152-172, very helpful paragraph, thank you for including. It would be good to summarize the findings from these studies, rather than just reporting on what the studies examined. Since the para focuses on Ethiopia, I suggest you drop Heckert et al. which was conducted in Burkina.

4. Methods - line 269, you only include married or co-habitating women, but then you control for marital status in you model. The variable in table 2 is called "married". What is the reference category? Co-habitating? Please clarify in the text.

5. Methods - lines 388-295, please define acceptable cut-offs for SRMR and CD, which you report.

6. Methods - in re-reading the paper I realized the authors have not addressed the issue of clustering. Typically DHS analyses account for clustering at the EA level.

7. Methods - sample size - you report that you restrict analyses to 10,641 women (line 267) but table 2 has fewer observations. Were observations dropped due to missing data? please describe in the methods.

8. Discussion - several times the authors say empowerment is important for "child development", but child development was not a measure included in the paper.

7. PLOS authors have the option to publish the peer review history of their article (what does this mean?). If published, this will include your full peer review and any attached files.

Reviewer #2: No

---

## [Author Response · Author response to Decision Letter 1]

16 Jun 2020

1. EFA/CFA - conducting EFA and CFA on the full sample is a bit unorthodox. Typically, one random split half is used for EFA, and the remaining half for CFA. It would be helpful if the authors provide further rationale or a reference for using the full sample for these analyses.

Response: 

The reviewer is right in noting that the conventional approach for implementing CFA and EFA is using a split sample. However, we refer some studies and the studies indicated that implementing the EFA and CFA analyses using a full sample and split sample leads to similar results as far as methodological explanations can account for cases in which EFA and CFA lead to different conclusions based on the same sample (see e.g., (Van Prooijen & Van Der Kloot, 2001). These methodological issues include inadequate evidence on the indicators to measure the component, conservativeness of the CFA model and inappropriate applications of CFA. We address the methodological issues in measuring the indicators related to women’s empowerment as follows: first, based on previous literature we operationalized women’s empowerment variables to check whether the indicators related to women’s empowerment variable represent an existing entity that can be categorized in the same underlying latent factors. For example, women’s participation in household decision making have four indicators. Before considering these indicators in one latent dimension, first we coded the responses for each indicator and dichotomized them to know if the women is empowered or not (see section 4.1.2 of the manuscript). Second, some of the indicators related to women’s empowerment, such as women’s socio-economic status indicators, are not known in literature. Therefore, we performed EFA first. Then CFA is performed. (see section 4.21 line 476-486) 

2. Supplemental table 1 - I suggest you move this table to the main text. Rereading the paper now, I find myself wanting to see a description of the sample.

Response: We have now added Table 1 to section 5 of the main manuscript. We also included an accompanying description of the table (see section 5.1). 

3. Introduction - thank you for including the additional literature. This is very helpful in putting your study in context. Lines 99-102, I agree that this is a limitation of the previous literature, but some the studies never aimed to assess infectious disease or causes of death, e.g. Heckert et al. aimed to asses only nutritional status. I suggest the authors re-phrase this sentence to more properly cite the literature. Lines 152-172, very helpful paragraph, thank you for including. It would be good to summarize the findings from these studies, rather than just reporting on what the studies examined. Since the para focuses on Ethiopia; I suggest you drop Heckert et al. which was conducted in Burkina.

Response: We agree with the reviewer that Heckert et al., 2019 considered only child nutritional outcome. Hence, we have dropped the reference. We have also (i) re-phrased the sentence and (ii) summarized and wrote in one paragraph the main findings with a new paragraph. 

4. Methods - line 269, you only include married or co-habitating women, but then you control for marital status in you model. The variable in table 2 is called "married". What is the reference category? Co-habitating? Please clarify in the text.

Response: In this study, we considered not only women who are currently married and living with partner but also other categories of marital status of such as widowed, divorced and separated. The variable marital status in our case is categorized as 1 if the women is married (stands for married or living with partner categories), 0 otherwise. In the DHS survey, women - either married or not married - responded to the questionnaire related to women’s empowerment variables and hence we use all women sample in our analysis. Brief description of this has been provided in the revised version of the manuscript (see section 4.1).

5. Methods - lines 388-295, please define acceptable cut-offs for SRMR and CD, which you report.

Response: Acceptable cut of value for SRMR is 0.08 or lower. The closer to 0, the better the fit of the model is (Hu & Bentler, 1999). A perfect correspond to Coefficient of Determination (CD) is 1. The closer the value of CD to 1, the better the goodness of fit of the model. Literatures indicated that the acceptable cut off value of CD is depending on the number of exogenous latent variables. Values of 0.67, 0.33 and 0.19 for endogenous latent variables in the inner path model are described as substantial, moderate and weak (Chin, 1998). (Henseler, Ringle, & Sinkovics, 2009) suggested that if endogenous latent variable explained by only a few exogenous latent variables, “Moderate” CD may be acceptable. If the endogenous latent variable relies on several exogenous latent variables, the CD value should exhibit at least a substantial level. We have now included this in text revised version of the manuscript (see section 4.2.1 line 542-550)

6. Methods - in re-reading the paper I realized the authors have not addressed the issue of clustering. Typically, DHS analyses account for clustering at the EA level.

Response: We have taken this reviewer’s comment and addressed the issue of clustering through clustering the standard errors at EA level. The result and the sign of the coefficient is not changed after considering clustering, but we presented the results with clustering (see Table 3). 

7. Methods - sample size - you report that you restrict analyses to 10,641 women (line 267) but table 2 has fewer observations. Were observations dropped due to missing data? please describe in the methods.

Response: Our final sample size for the analysis depends on the children’s health outcome variables. So, the lower observation in some of the models is due to missing values (observations) for the outcome variable under consideration. We have not explained this point in the manuscript (as a note to Table 3) to make it clear to the reader.

8. Discussion - several times the authors say empowerment is important for "child development", but child development was not a measure included in the paper.

Response: Point taken. We replaced the word “child development” with appropriate outcome variables throughout the paper. 

References 

Chin, W. W. (1998). The partial least squares approach to structural equation modeling. Modern Methods for Business Research, 295(2), 295–336.

Henseler, J., Ringle, C. M., & Sinkovics, R. R. (2009). The use of partial least squares path modeling in international marketing. In New challenges to international marketing. Emerald Group Publishing Limited.

Hu, L., & Bentler, P. M. (1999). Cutoff criteria for fit indexes in covariance structure analysis: Conventional criteria versus new alternatives. Structural Equation Modeling: A Multidisciplinary Journal, 6(1), 1–55.

Van Prooijen, J.-W., & Van Der Kloot, W. A. (2001). Confirmatory analysis of exploratively obtained factor structures. Educational and Psychological Measurement, 61(5), 777–792.

---

## [Decision Letter · Decision Letter 2]

24 Jun 2020

Associations between Women’s Empowerment and Children's Health Status in Ethiopia

PONE-D-20-00349R2

Dear Dr. Abreha,

We’re pleased to inform you that your manuscript has been judged scientifically suitable for publication and will be formally accepted for publication once it meets all outstanding technical requirements.

Kind regards,

Mellissa H Withers, PhD, MHS

Academic Editor

PLOS ONE

Additional Editor Comments (optional):

Reviewers' comments:

Reviewer's Responses to Questions

**Comments to the Author**

1. If the authors have adequately addressed your comments raised in a previous round of review and you feel that this manuscript is now acceptable for publication, you may indicate that here to bypass the “Comments to the Author” section, enter your conflict of interest statement in the “Confidential to Editor” section, and submit your "Accept" recommendation.

Reviewer #2: All comments have been addressed

2. Is the manuscript technically sound, and do the data support the conclusions?

Reviewer #2: (No Response)

3. Has the statistical analysis been performed appropriately and rigorously? 

Reviewer #2: (No Response)

4. Have the authors made all data underlying the findings in their manuscript fully available?

Reviewer #2: (No Response)

5. Is the manuscript presented in an intelligible fashion and written in standard English?

Reviewer #2: (No Response)

6. Review Comments to the Author

Reviewer #2: (No Response)

7. PLOS authors have the option to publish the peer review history of their article (what does this mean?). If published, this will include your full peer review and any attached files.

Reviewer #2: No

---

## [Editor Report · Acceptance letter]

7 Jul 2020

PONE-D-20-00349R2 

Associations between Women’s Empowerment and Children's Health Status in Ethiopia 

Dear Dr. Abreha:

I'm pleased to inform you that your manuscript has been deemed suitable for publication in PLOS ONE. Congratulations! Your manuscript is now with our production department. 

Kind regards, 

on behalf of

Dr. Mellissa H Withers 

Academic Editor

PLOS ONE